# Real-Time Stress Analysis Affecting Nurse during Elective Spinal Surgery Using a Wearable Device

**DOI:** 10.3390/brainsci12070909

**Published:** 2022-07-12

**Authors:** Sayhyun Sung, Ji-Won Kwon, Jung-Eun Kim, Yu-Jin Lee, Soo-Bin Lee, Seung-Kyu Lee, Seong-Hwan Moon, Byung Ho Lee

**Affiliations:** 1Department of Orthopedic Surgery, College of Medicine, Ewha Womans University, Seoul 07804, Korea; sahyunsung@ewha.ac.kr; 2Department of Orthopedic Surgery, College of Medicine, Yonsei University, Seoul 03722, Korea; kwonjjanng@yuhs.ac (J.-W.K.); terry1376@yuhs.ac (S.-K.L.); shmoon@yuhs.ac (S.-H.M.); 3Division of Nursing, Severance Hospital, Seoul 03722, Korea; jekim11@yuhs.ac (J.-E.K.); yjice@yuhs.ac (Y.-J.L.); 4Department of Orthopedic Surgery, College of Medicine, Catholic-Kwandong University, Incheon 25601, Korea; sumanzzz@ish.ac.kr

**Keywords:** electroencephalography, heart rate variability, intraoperative stress, spine surgery, scrub nurse, circulating nurse

## Abstract

Successful spinal surgery demands high levels of concentration and cooperation from participating health care workers. The intraoperative stress levels and concentration levels of surgeons have been studied previously; however, those of nurses are rarely studied. Therefore, the purpose of this study is to understand the stresses affecting surgical nurses by their participating role during spinal surgery. A total of 160 surgical stress records were obtained during 40 surgeries, including electroencephalography (EEG) signals and heart rate variability (HRV) from three orthopedic spinal surgeons and six nurses; concentration, tension level and physical stress were analyzed. Levels of both concentration and tension were significantly higher in circulating nurses during all surgical stages (*p* < 0.05). Both beats per minute and low frequency/high frequency ratios, which reflect physical stress, were higher in scrub nurses (*p* < 0.05). As the surgical experience of scrub nurses increased, the key parameters related to stress tended to decrease (*p* < 0.01). These results will contribute to understanding the pattern of intraoperative stress of surgical nurses, and therefore help in enhancing the teamwork of the surgical team for optimal outcomes.

## 1. Introduction

Stress and burn out of physicians and other healthcare providers can adversely affect patient outcomes [1]. Especially in spinal surgery which demands high levels of concentration and cooperation during a long duration of time, extreme stress among surgeons and medical staff can not only lead to dissatisfactory surgical outcomes but can also cause serious adverse events [2,3]. The impact of stress on surgeons has been investigated using questionnaires or through measuring heart rate (HR), HR variability (HRV), thermal activity, sympathovagal balance, or stress biomarker levels such as salivary cortisol [4,5,6]. However, these methods cannot distinguish emotional stress from physical stress and most of the measurements were conducted post-surgery and not intraoperatively. Some studies described the effects of stress on teamwork performance and the stresses present within operating rooms [7]. Results from a questionnaire survey revealed low response rates as well as the under- or overstating of work experience based on the relationship between the interviewer and the interviewee [8,9]. Advances in the technology and utilization of wearable devices have allowed researchers to obtain intraoperative stress data, including electroencephalography (EEG) and HRV data, in real time, enabling the evaluation of physical and mental stress results simultaneously [10,11]. These studies showed that surgeons’ stress levels increased during surgery compared to the resting state, and the less experienced the surgeon, the longer the operation time, and the higher the blood loss, the more the stress levels of the surgeons increased [10,11].

Stress levels and their effect on medical staff other than surgeons are rarely studied [12,13,14]. Therefore, we aimed to analyze the intraoperative stress levels of surgical nurses, which can help us understand the hardships of the participating medical staff other than the surgeons and, thus, help in improving the teamwork of the surgical team. The intraoperative stress parameters of participating doctors and surgical nurses were analyzed using wearable devices. These real-time data were used to analyze whether the stress patterns differed according to work experience and participating roles [12].

## 2. Materials and Methods

From June 2019 to December 2019, 160 surgical stress records, including intraoperative EEG signals and HRV data, were obtained during 40 spinal surgeries at a tertiary hospital. The operator, assistants, scrub nurses, and circulating nurses who participated in each surgery all wore the wearable device for data collection. A total of 160 stress records were obtained from the participating medical staff.

The operator participated in all stages of the operation as the main surgeon, and the first assistant surgeon assisted the operator throughout the surgery. The scrub nurse prepared the instruments and materials and handed them directly to the surgeon. The circulating nurse checked for the overall process, prepared additional instruments not available in the surgical field, and aided the scrub nurses.

The work experience of the orthopedic surgeons ranged from 1 to 8 years, and the work experience of the nurses ranged from 1 to 21 years. Demographics, including subspecialties, are presented in Table 1.

The operative time and the amount of blood loss were analyzed for all cases. The surgeries of degenerative lumbar spine disease were selected for stress standardization, and emergency trauma cases were excluded. The surgeries included degenerative spondylolisthesis (15 cases), lumbar spinal stenosis (14 cases), adjacent segment disease (5 cases), spondylolytic spondylolisthesis (3 cases), and degenerative scoliosis (3 cases). The stress levels during the surgery and at rest were measured and compared.

All the included patients had undergone decompression and fusion surgeries [15,16]. For the analysis of stress levels according to surgical stage, all spinal surgeries were divided into the following stages: (1) incision (and dissection), (2) decompression (including decompressive laminectomy and discectomy), (3) screw fixation-rod assembly, and (4) closure, as described previously [10].

Other factors such as a poorly functioning scrub team and interruptions during surgery are believed to contribute to the intraoperative stress of the team [4,12,17]. However, the studied surgical team had at least 6 months of work experience and performed more than 200 spinal surgeries together; therefore, it is unlikely that the poor teamwork would have contributed to intraoperative stress levels [18]. None of the enrolled surgeons took beta-blockers or anti-arrhythmic medications that could affect their HRV values.

### 2.1. Evaluation of Intraoperative Stress Using a Wearable Device

The wearable EEG device used in this study was previously described in a psychiatric study and an orthopedic study, which examined the effects of post-traumatic stress disorder on EEG patterns and intraoperative stress on orthopedic spinal surgeons [10,19].

The two-channel EEG device (model: Amp GS5001; SOSO H&C, Kyungpook University, Daegu, Korea) measured the cortical activity of the frontal lobe for 3 min while the participant was at rest. A headband was worn to secure the dry electrodes to the Fp1 and Fp2 sites according to the 10–20 system. The reference electrode was placed on the right earlobe. The sampling rate of 256 Hz and online notch filtering (60 Hz) was applied. The data were inspected for artifacts before preprocessing the EEG data. A photoplethysmography sensor was placed in the right ear with a reference EEG sensor to continuously monitor the HRV [20].

MATLAB 2012 software (MathWorks, Inc., Natick, MA, USA) was used to analyze the EEG data. Amplitudes were calculated using a fast Fourier transform algorithm with a bandpass filter of 1–50 Hz. The frequency power was calculated as the square of the amplitudes, and the signals were translated as the following: relative delta (1–4 Hz), theta (4–8 Hz), alpha (8–12 Hz), low beta (12–15 Hz), mid-range beta (M-beta, 15–20 Hz), high beta (H-beta, 20–30 Hz), and gamma signals (30–50 Hz). Relative frequency power was converted to a percentage of the whole frequency and provides information about interactions between spectral oscillations in each band. Artifacts exceeding ±100 μV were excluded at all electrode sites. Thirty artifact-free epochs (epoch length, 2.048 s) were collected for each subject.

The device was worn preoperatively, and the measurements were taken from the time of handwashing for the surgeons and preparation of the operating table for the nurses until the end of the surgery. The total measurement time was about 20 min longer in nurses than in surgeons. The real-time stress data were saved every 10 min in separate files, and later merged into a single file. The data were analyzed according to the surgical stage, and compared with the level detected while at rest. The exact time and duration of each surgical stage (i.e., incision, decompression, screw fixation, and closure) were found in the medical records. The HR and EEG signal at rest were measured for 3 min, and these baseline data were used to compare the intraoperative data.

Alpha, delta and theta waves appear when the person is relaxed, drowsy, idling or during meditation [21,22]. Beta and gamma waves can be observed during active thinking tasks. Low beta waves are related to active, busy or anxious thinking and active concentration. M-beta waves are associated with increases in energy, concentration, anxiety and performance. High beta waves are associated with significant stress, anxiety, paranoia, and high arousal [23,24,25]. Gamma waves appear when carrying out multiple cognitive functions [24]. Increased sensorimotor rhythm (SMR) is associated with a reduction in commission errors and improved cognitive function [26]. In regard to HRV-related measures, an increased low frequency/high frequency (LF/HF) ratio is associated with high stress levels [27].

Concentration is calculated by dividing the sum of SMR and M-beta waves by theta waves. Tension is calculated by dividing the H-beta wave by the alpha wave [28]. Because the collected data using the wearable device in this study had different measurement units, the company SOSO H & C suggested the following formula to calculate the concentration and tension [10,11,29,30,31,32,33]: Concentration = [[LOG10[(SMR + M-Beta)/Theta] + 1.8]/2.8] × 99 + 1
Tension = [[LOG10(H-Beta/Alpha) + 1.0843]/2.058993] × 99 + 1

HRV was determined by measuring beats per minute (BPM), root mean square of successive differences, LF band, and HF band using an ear probe. Then, BPM and the LF/HF ratio, which is associated with levels of surgical stress, were analyzed.

### 2.2. Statistical Analyses

The independent Student’s *t*-test and analysis of variance (ANOVA) were used to analyze the data collected from the study participants. A non-parametric Kruskal–Wallis test was used to identify the differences in stress data according to work experiences and participating roles of surgical nurses (circulating or scrub). A bivariate Pearson correlation analysis was performed to evaluate the correlation between intraoperative stress levels and possible influencing factors, such as surgical experience, intraoperative blood loss, operative time and surgeon stress level.

All statistical analyses were performed using the SPSS 22.0 statistical package (SPSS, International Business Machines Corp., New York, NY, USA). *p* values less than 0.05 were considered statistically significant.

This study was approved by the institutional review board (IRB) of the authors’ hospital (IRB No. 4-2018-0984).

## 3. Results

### 3.1. Stress Parameters Differed between Surgeons, Assistants, Scrub Nurses, and Circulating Nurses

All parameters, including EEG (delta, theta, alpha, SMR, M-beta, H-beta, and gamma waves), HRV (BPM and LF/HF ratio), and levels of concentration and tension, significantly differed by role during surgery (ANOVA, *p* < 0.05, respectively). The key parameters are shown in Figure 1.

Additionally, Bonferroni post hoc analyses were performed to confirm the differences between groups. During the incision stage, the levels of concentration and tension were highest for surgeons. BPM values of surgeons and scrub nurses were higher than those of assistants. The LF/HF ratio was highest in scrub nurses but lowest in circulating nurses during the incision stage (*p* < 0.05).

During the instrumentation stage, the levels of concentration and tension were highest in surgeons, followed by circulating nurses; these parameters were lowest in scrub nurses (*p* < 0.05). BPM values were highest in surgeons but lowest in circulating nurses, with no differences observed between assistants and scrub nurses. The LF/HF ratio was lowest in circulating nurses (*p* < 0.05).

During the decompression stage, the levels of concentration and tension were highest in surgeons, followed by circulating nurses. The LF/HF ratio was highest in surgeons and lowest in circulating nurses. BPM values were highest in surgeons, followed by scrub nurses (*p* < 0.05).

During the closure stage, the levels of concentration were highest in circulating nurses, followed by surgeons. Levels of tension were highest in surgeons, followed by circulating nurses. BPM values were highest in surgeons, followed by scrub nurses. The LF/HF ratio was highest in scrub nurses (*p* < 0.05).

Other stress parameters (e.g., M-beta, H-beta, gamma, and SMR waves) were highest in surgeons, followed by circulating nurses, and were lowest in scrub nurses during all stages of surgery. In contrast, the parameters reflecting a relaxed state, such as delta, theta, and alpha waves, were highest in scrub nurses and assistants during the incision stage, whereas alpha waves were highest in surgeons. Both delta and theta waves during the instrumentation, decompression, and closure stages were highest in scrub nurses (*p* < 0.05).

### 3.2. All Parameters Significantly Differed between Scrub and Circulating Nurses during All Surgical Stages

The levels of concentration and tension were significantly higher in circulating nurses than in scrub nurses. However, BPM values and the LF/HF ratio, which reflect physical stresses, were higher in scrub nurses than in circulating nurses. All parameters related to a relaxed state were significantly higher in scrub nurses during all surgical stages compared to circulating nurses. The levels of concentration and other stress-related factors, including M-beta, H-beta, and gamma waves, were also significantly higher in circulating nurses during all surgical stages compared to scrub nurses (independent Student’s *t*-test, *p* < 0.05) (Figure 2; Figure 3).

The levels of concentration and tension, BPM, and LF/HF ratio levels of nurses during all surgical stages. All parameters were significantly different between scrub and circulating nurses (*p* < 0.05, Student’s *t*-test) (Unit: percentage). The levels of concentration and tension were significantly higher in circulating nurses than in scrub nurses. However, BPM values and the LF/HF ratio, which reflect physical stresses, were higher in scrub nurses than in circulating nurses.

During all surgical stages, theta, delta, and alpha waves were significantly different between scrub and circulating nurses (*p* < 0.05, Student’s *t*-test). The levels of concentration and other stress-related factors, including M-beta, H-beta, and gamma waves, were also significantly higher in circulating nurses during all surgical stages compared to scrub nurses (independent Student’s *t*-test, *p* < 0.05).

### 3.3. Stress Parameters Differed in Scrub Nurses Based on Work Experience

More experienced scrub nurses demonstrated lower levels of stress during surgery (Figure 4). All parameters significantly differed between groups during surgery (Kruskal–Wallis, *p* < 0.05).

### 3.4. Correlation of Stress Parameters Differed by Role

Overall, as work experience increased, the relaxation-related parameters (e.g., delta, theta, and alpha waves) correlated positively, but the concentration-related factors (e.g., SMR, M-beta, H-beta, and gamma waves; levels of concentration and tension; and LF/HF ratio) were negatively correlated (Pearson correlation analyses, *p* < 0.01) (Table 2).

A longer duration of surgery positively correlated with delta and gamma waves and BPM values in scrub nurses. However, the SMR, M-beta, and H-beta waves; levels of tension; and LF/HF ratio correlated negatively with the duration of surgery, suggesting that scrub nurses became tired or lost focus during long surgeries. Similar patterns were observed in circulating nurses (Pearson correlation analyses, *p* < 0.01).

The volume of intraoperative bleeding positively correlated with the delta, SMR, M-beta, H-beta, and gamma waves; levels of tension; BPM values; and LH/HF ratio in scrub nurses. However, theta and alpha waves correlated negatively with bleeding volume, indicating that scrub nurses were highly focused and under greater stress during surgeries with high volumes of intraoperative bleeding. Similar patterns were observed in circulating nurses (Pearson correlation analyses, *p* < 0.01).

The level of stress in surgeons negatively correlated with those of nurses, but differed between scrub nurses and circulating nurses (Pearson correlation analyses, *p* < 0.01).

## 4. Discussion

This study is part of a series to investigate the effects of real-time stress on surgeons and OR personnel during the entire intraoperative period [10,11]. There are various factors in the operating room (OR) that could increase the risk of surgery, including the patient’s condition, the surgeon’s surgical proficiency, teamwork and environmental conditions [17,34,35,36,37]. These can all contribute to the participating medical staff’s stress levels during the surgery.

In our study, all stress markers increased during surgery, although the pattern differed based on the role of the individual. Surgeons were under maximal stress, but scrub nurses appeared to have more physical stress because they spend more of their time standing and moving complex and heavy instruments. Orthopedic spinal surgery is a physically demanding procedure, owing to the use of a hammer, drill, and screw-rods, which requires strong skeletal muscles [38]. These observations were also reported in previous papers on orthopedic surgeons performing spinal surgeries [10,11].

Circulating nurses were under significant mental stress during surgery, as they must continuously concentrate to prevent unexpected events and coordinate the needs of the surgical team [7]. For experienced and focused circulating nurses, stress-related EEG signals and HRV values were observed even though there was no apparent increase in HR compared to scrub nurses. This finding may be explained by the fact that circulating nurses with sufficient experience in orthopedic spinal surgery can anticipate possible adverse events and immediately respond to them during the surgical procedure. This experience maintains their concentration during every step of the surgical procedure.

The decompression stage of spinal surgery is considered to be the most difficult part for surgeons because of the handling and manipulation of the neural tissue, which, if damaged, could lead to neurological deficits such as paralysis or neuropathic pain [10,15]. However, for scrub nurses, no stage-specific patterns of stress were observed. Rather, when the surgery reached the closure stage, the levels of concentration and tension decreased dramatically and only the physical stress remained.

After the main procedure, the levels of concentration and tension and physical stress parameters decreased in surgeons, assistants, and scrub nurses. However, in circulating nurses, the levels of concentration and stress-related parameters increased, likely because of their job tasks (e.g., counting the materials used, wrapping up the used instruments, and preparing for the next surgery).

Another aspect that should be highlighted in this study is the physical stress observed in terms of increased HR and LF/HF ratios during surgery. Orthopedic surgery is physically demanding for all surgical team members [38]. Because the scrub and circulating nurses prepare and arrange the OR and pass the instruments to the surgeons, they endure high levels of physical stress during the surgery. Some orthopedic nurses have been found to experience musculoskeletal problems because of their work [39], so physical stress as a contributing factor should not be underestimated.

The longer operation time correlated with tiredness and loss of concentration. Additionally, a higher intraoperative bleeding volume correlated positively with higher stress levels in both scrub nurses and circulating nurses. Stress levels were especially higher in the circulating nurses than in the scrub nurses. This is probably because in a bleeding situation, the circulating nurses were often called suddenly, and they were busy preparing other instruments or coagulants for hemostasis.

Since the study included surgical cases performed in a tertiary hospital, the surgery is more likely to be challenging and the operative time is likely to be longer compared to other institutions. There are also additional personnel including training residents and fellows and this could affect the nurses’ stress levels; the working environment is likely to be different compared to working with a proficient leading surgeon. Another limitation of this study is the small number of study subjects. However, this is the first study to report the intraoperative stresses affecting orthopedic nurses in the OR based on their role and occupational experiences.

## 5. Conclusions

For orthopedic surgical nurses, high levels of mental and/or physical stress persist throughout the surgery, depending on their specific role. Scrub nurses tended to suffer more physical stress, while the circulating nurses had high levels of tension and concentration during the entire surgery. These results help in understanding the stress patterns of the medical staff other than the surgeons, which can contribute to better teamwork. This will lead to better surgical outcomes and help prevent burn out among medical staff.

## Figures and Tables

**Figure 1 brainsci-12-00909-f001:**
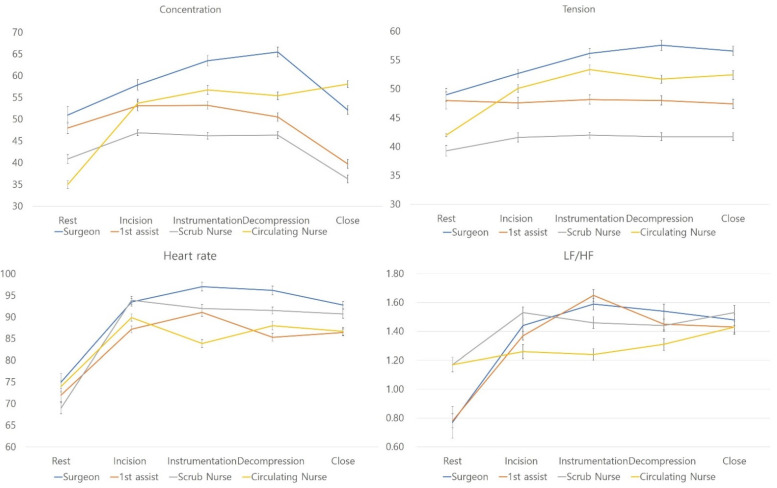
Stress-related parameters during surgery.

**Figure 2 brainsci-12-00909-f002:**
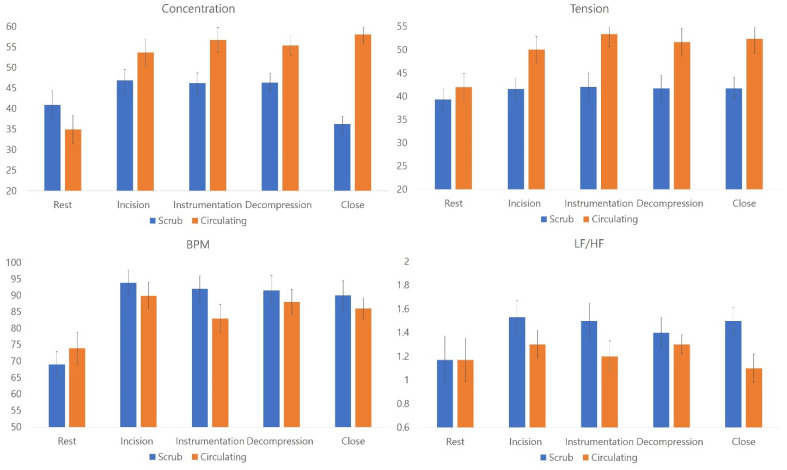
Comparison of stress-related parameters between scrub nurses and circulating nurses.

**Figure 3 brainsci-12-00909-f003:**
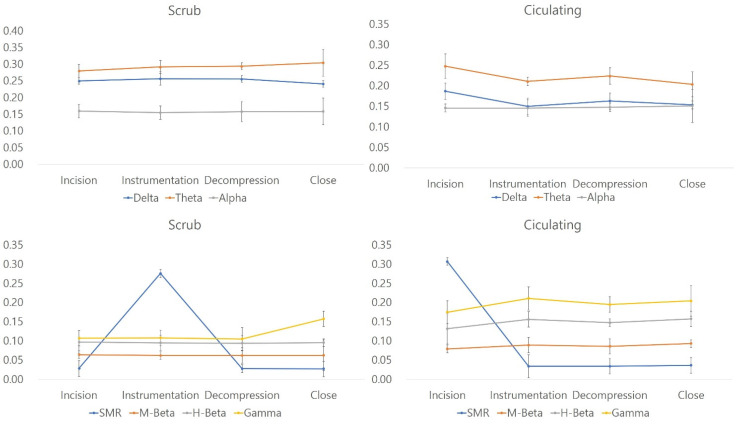
Analyses of EEG waves between scrub and circulating nurses.

**Figure 4 brainsci-12-00909-f004:**
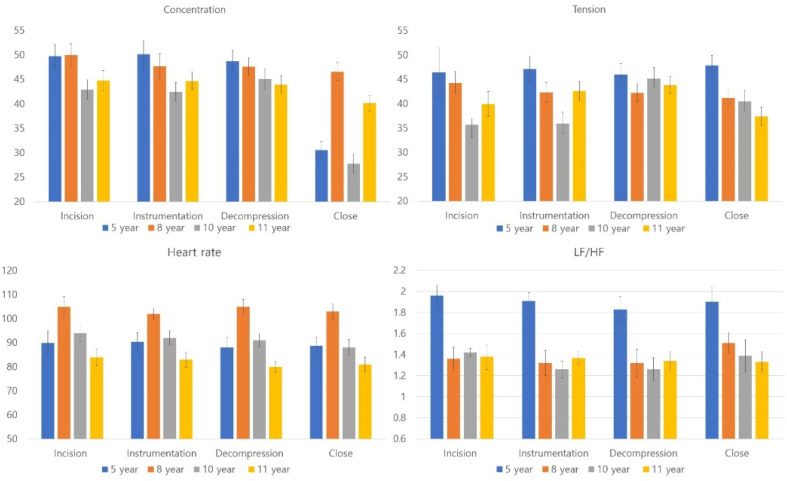
Stress-related parameters based on the work experience of scrub nurses.

**Table 1 brainsci-12-00909-t001:** Demographic data of enrolled study participants.

	Surgeon			Nurse					
Role during Surgery	Operator	Assistant	Assistant	Scrub	Scrub	Scrub	Scrub	Circulating	Circulating
Subject ID	O1	A1	A2	S1	S2	S3	S4	C1	C2
Age (years)	43	35	36	35	34	32	29	45	44
Experience as a spinal surgeon/operation room nurse (years)	9	1 (first-yearclinical fellow)	1 (first-yearclinical fellow)	11	10	8	5	22	19
Experience in orthopedic surgery (years)	16	8	8	3	10	5	3	17	16
Accumulated number of orthopedic surgeries (cases)				3600 cases	5000 cases	4800 cases	3000 cases	Over 15,000 cases	Over 10,000 cases
Accumulated surgical experiences in spinal surgery (cases)	1700 cases as an operator	150 cases as an assistant	100 cases as an assistant	500 cases	2500 cases	400 cases	240 cases	Over 7000 cases	Over 4000 cases
Enrolled cases in the present study	40	20	20	10	10	10	10	20	20

**Table 2 brainsci-12-00909-t002:** Correlation of stress-related parameters between scrub and circulating nurses. ** *p* < 0.01, Pearson correlation analyses.

		Delta Waves	Theta Waves	Alpha Waves	SMR Waves	M-Beta Waves	H-Beta Waves	Gamma Waves	Concentration Level	Tension Level	BPM	LF/HF Ratio
Experience as OR nurse (Yrs)	Scrub		0.925 **	0.447 **	−0.593 **	−0.659 **	−0.767 **	−0.787 **	−0.227 **	−0.832 **	−0.287 **	−0.871 **
	Circulating	0.910 **	0.858 **	−0.782 **	−0.782 **	−0.742 **	−0.889 **	−0.907 **	−0.833 **	−0.900 **	0.309 **	
Experience in orthopedic surgery (Yrs)	Scrub	0.485 **	0.298 **	0.388 **		−0.358 **	−0.583 **	−0.718 **	−0.312 **	−0.662 **	0.250 **	−0.482 **
	Circulating	0.910 **	0.858 **	−0.782 **	−0.742 **	−0.834 **	−0.889 **	−0.907 **	−0.833 **	−0.900 **	0.309 **	
Experience in orthopedic surgery (cases)	Scrub		0.452 **	0.763 **			−0.429 **	−0.652 **		−0.655 **	0.599 **	−0.748 **
	Circulating	0.910 **	0.858 **	−0.782 **	−0.742 **	−0.834 **	−0.889 **	−0.907 **	−0.833 **	−0.900 **	0.309 **	
Experience in spinal surgery (cases)	Scrub	0.625 **	0.380 **	0.234 **	−0.375 **	−0.539 **	−0.710 **	−0.780 **	−0.404 **	−0.718 **		−0.460 **
	Circulating	0.910 **	0.858 **	−0.782 **	−0.742 **	−0.834 **	−0.889 **	−0.907 **	−0.833 **	−0.900 **	0.309 **	
Duration of surgery	Scrub		0.272 **	−0.760 **	−0.498 **	−0.263 **		0.268 **		0.285 **	−0.893 **	0.508 **
	Circulating	0.908 **	0.888 **	−0.853 **	−0.838 **	−0.891 **	−0.903 **	−0.892 **	−0.884 **	−0.891 **		
Intraoperative bleeding	Scrub	0.214 **	−0.716 **	−0.189 **	0.473 **	0.391 **	0.342 **	0.271 **		0.359 **	0.445 **	0.538 **
	Circulating	−0.895 **	−0.878 **	0.849 **	0.837 **	0.885 **	0.892 **	0.877 **	0.877 **	0.876 **		
Stress of surgeon	Scrub	0.894 **	0.902 **	−0.028 **	−0.869 **	−0.889 **	−0.898 **	−0.907 **	−0.717 **	−0.861 **		
	Circulating	0.441 **	0.484 **	−0.607 **	−0.430 **	−0.411 **	−0.442 **	−0.427 **	−0.218 **	−0.345 **	−0.523 **	−0.611 **

## Data Availability

The data presented in this study are available upon request from the corresponding author.

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
