# Peer review of "Real-Time Stress Analysis Affecting Nurse during Elective Spinal Surgery Using a Wearable Device"

_brainsci, 2022, doi:10.3390/brainsci12070909_

Round 1

Reviewer 1 Report

The authors presented a study on real-time stress analysis of nurses during surgeries which is very important as such medical staff feels more mental pressure while conducting patient operations. Before proceeding further, I would like to propose some comments for the authors to make it more comprehensive

  1. The abstract should restructure with clear aim and study findings with possible values
  2. The introduction section is poorly written with a limited literature, missing research gaps, and the importance of the present study
  3. Is the dataset publicly available?  If yes mention the source. If not has participant consent is majorly missing
  4. The data is presenting in this work is 2019 is there any reason behind considering the particular time zone
  5. What is the importance of name initials in Table 1? Instead, authors can add subject ID
  6. Figure 1 is not clear to observe
  7. The conclusion section should come with generic outcomes and future recommendations rather than specific objectives
  8. The manuscript needs English revision under the supervision of native speakers   

Author Response

Response to Reviewer 1 Comments

We are very grateful for the opportunity to revise this manuscript. We thank you for taking your time and effort to provide helpful and constructive comments. We have provided point-by-point answers to the specific comments raised by the reviewers and the modifications have been applied to the new “revised manuscript”. We hope the revised revision of the manuscript will meet the requirements for publication.

  1. The abstract should restructure with clear aim and study findings with possible values

Response: Thank you for the comment. We made changes in the abstract to clarify our objective and results.

Page 1, Abstract)

Abstract: Successful spinal surgery demands high levels of concentration and cooperation of participating health care workers. The intraoperative stress levels and concentration of the surgeons have been studied previously, however those of the nurses are rarely studied. Therefore, the purpose of this study is to understand the stresses affecting surgical nurses by their participating role during spinal surgery. A total of 160 surgical stress records were obtained during 40 surgeries, including electroencephalography (EEG) signals and heart rate variability (HRV) from three orthopedic spinal surgeons and six nurses and concentration, tension level and physical stress were analyzed. Both levels of concentration and tension were significantly higher in circulating nurses during all surgical stages (P < 0.05). Both beats per minute and low frequency/high frequency ratios, which reflect physical stress, were higher in scrub nurses (P < 0.05). As the surgical experiences of scrub nurses increased, the key parameters related to stress tended to decrease (P<0.01). These results will contribute to understanding the pattern of intraoperative stress of the surgical nurses, and therefore help in enhancing the teamwork of the surgical team for optimal outcome.

  1. The introduction section is poorly written with a limited literature, missing research gaps, and the importance of the present study

Response:  We decided to study the stress levels of participating surgical team workers, not only surgeons as previously studied. We hypothesized that the stress levels would be different according to the participating roles, and that the levels of stress could also be influence by personal factors such as experience. We rewrote the introduction to clarify the purpose of this study, and added more references on the stress of the healthcare personnel.

Page 1-2, introduction)

Stress and burn out of the physicians and other healthcare providers can adversely affect the patient outcome [1]. Especially in spinal surgery which demands high levels of concentration and cooperation during a long duration of time, extreme stress of the surgeons and the medical staff can not only lead to dissatisfactory surgical outcome but also can cause serious adverse event [2, 3]. The impact of stress on surgeons have been investigated using questionnaires or measuring heart rate (HR), HR variability (HRV), thermal activity, sympathovagal balance, or stress biomarker levels such as salivary cortisol [4-6]. However, these methods cannot distinguish emotional stress from physical stress and most of the measurements were conducted post-surgery and not intraoperatively. Some studies described the effects of stress on teamwork performance and the stresses present within operating rooms [7]. Results from a questionnaire survey revealed low response rates as well as the under- or overstating of work experience based on the relationship between the interviewer and the interviewee [8, 9]. Advances in the technology and utilization of wearable devices have allowed researchers to obtain intraoperative stress data, including electroencephalography (EEG) and HRV data, in real time, enabling the evaluation of physical and mental stress results simultaneously [10, 11]. These studies showed that surgeon’s stress levels increased during surgery compared to the resting state, and the less experienced the surgeon, the longer the operation time, and the higher the blood loss, the stress levels of the surgeons increased [10, 11].

The stress levels and its effect of medical staff other than the surgeons are rarely studied [12-14]. Therefore, we aimed to analyze the intraoperative stress levels of the surgical nurses, which can help understand the hardships of the participating medical staff other than the surgeons and thus help in improving the teamwork of the surgical team. The intraoperative stress parameters of participating doctors and surgical nurses were analyzed using wearable devices. These real time data were used to analyze whether the stress patterns differed according to work experience and participating roles [12].

  1. Is the dataset publicly available?  If yes mention the source. If not has participant consent is majorly missing

Response: The data we used in this study is automatically recorded data in the memory card of the wearable device (model: Amp GS5001; SOSO H&C, Kyungpook University, Daegu, Korea). The data will be provided upon appropriate request. The informed consent of the medical staff who participated in this study was obtained before data collection.

  1. The data is presenting in this work is 2019 is there any reason behind considering the particular time zone

Response: This is a part of a series of studies we performed on the stress levels of the medical staff during spinal surgeries. We first obtained real time analysis data on the surgeons in 2018, and we started obtaining data from the scrub nurses and patients. The data we collected in 2018 was published in two studies in 2020 and 2021.

J.-W. Kwon, S. Sung, S.-B. Lee, H.-M. Lee, S.-H. Moon, and B. H. Lee, "Intraoperative real-time stress in degenerative lumbar spine surgery: simultaneous analysis of electroencephalography signals and heart rate variability: a pilot study," The Spine Journal, vol 20, no. 8, pp. 1203-1210, 2020.

J.-W. Kwon, S.-B. Lee, S. Sung et al., "Which factors affect the stress of intraoperative orthopedic surgeons by using electroencephalography signals and heart rate variability?," Sensors, vol 21, no. 12, p. 4016, 2021.

  1. What is the importance of name initials in Table 1? Instead, authors can add subject ID

Response: We agree with your opinion. We changed the name initials to subject ID.

  1. Figure 1 is not clear to observe

Response: We checked the resolution of the figure, and change to the one with higher definition.

  1. The conclusion section should come with generic outcomes and future recommendations rather than specific objectives.

Response: We agree that the previous conclusion consisted of specific results. We edited the conclusion as you suggested.

Page 13, Conclusion)

For orthopedic surgical nurses, high levels of mental and/or physical stress persist throughout the surgery, depending on their specific role. Scrub nurses tend to suffer more physical stress, while the circulating nurses had high levels of tension and concentration during the entire surgery. These results help understanding the stress patterns of the medical staff other than the surgeons, which can contribute to a better teamwork. This will lead to better surgical outcome and help prevent burn out of the medical staff.

  1. The manuscript needs English revision under the supervision of native speakers   

Response: The text had undergone official English editing by a professional English editing service prior to submission. We attached the certificate of editing. We further checked for grammatical errors after revising the manuscript.

Reviewer 2 Report

This paper examines levels of mental and physical stress suffered by surgeons and nruses during elective spinal surgeries. The topic is interesting and important, and the resource of subjects has high strengths. However, the manuscript was not well-written and the sections of methods and results were not clear as well. In this case, it hard to evaluate the results and the conclusion. 

1. Introduction

    The introduction is too simple and superficial. The authors may discuss the importance of the topic more clearly and introduce the previous studies in more details. It is beneficial to understand the past findings and then the contributions of the present study would be clear. In the current version, the authors mainly discussed the methodologies; even though, compared with the previous studies, the present research only added EEG and the merits were not introduced clearly.

2. Methods

    The Methods part is vague in the current version. For instance, what are the compositions of the 160 surgical stress records? Especially, the authors discussed the role differences later, but the detailed information about the role-relevant data was not clearly reported. The formulas calcuating concentration and tension were not clearly introduced. 

3. Results

    The analysis methods were also not clearly introduced. In addition, since the numbers of the subjects were limited, the sample size was very small, in this case, the t-tests and ANOVAs may be not appropriate. Even though, the authors did not report the details of the statistical analyses.

4. Dicussion

    The authors mentioned "team dynamics" many times when discussing the importance of the research topic and their findings. However, that is the definition of "team dynamics", which results supported the discussion of team dynamics?

Author Response

Response to Reviewer 2 Comments

We are very grateful for the opportunity to revise this manuscript. We thank you for taking your time and effort to provide helpful and constructive comments. We have provided point-by-point answers to the specific comments raised by the reviewers and the modifications have been applied to the new “revised manuscript”. We hope the revised revision of the manuscript will meet the requirements for publication.

  1. Introduction

    The introduction is too simple and superficial. The authors may discuss the importance of the topic more clearly and introduce the previous studies in more details. It is beneficial to understand the past findings and then the contributions of the present study would be clear. In the current version, the authors mainly discussed the methodologies; even though, compared with the previous studies, the present research only added EEG and the merits were not introduced clearly.

Response: Thank you for your comment. We edited introduction to include results of the previous studies as you suggested. The purpose of this study was to detect stress levels of the surgical team including the nurses, not only surgeons as many previous studies have observed. We emphasized that our study utilized real time analysis using wearable devices, compared to other studies which used questionnaire or means that could not differentiate physical and emotional stress.

Page 1-2, introduction)

Stress and burn out of the physicians and other healthcare providers can adversely affect the patient outcome [1]. Especially in spinal surgery which demands high levels of concentration and cooperation during a long duration of time, extreme stress of the surgeons and the medical staff can not only lead to dissatisfactory surgical outcome but also can cause serious adverse event [2, 3]. The impact of stress on surgeons have been investigated using questionnaires or measuring heart rate (HR), HR variability (HRV), thermal activity, sympathovagal balance, or stress biomarker levels such as salivary cortisol [4-6]. However, these methods cannot distinguish emotional stress from physical stress and most of the measurements were conducted post-surgery and not intraoperatively. Some studies described the effects of stress on teamwork performance and the stresses present within operating rooms [7]. Results from a questionnaire survey revealed low response rates as well as the under- or overstating of work experience based on the relationship between the interviewer and the interviewee [8, 9]. Advances in the technology and utilization of wearable devices have allowed researchers to obtain intraoperative stress data, including electroencephalography (EEG) and HRV data, in real time, enabling the evaluation of physical and mental stress results simultaneously [10, 11]. These studies showed that surgeon’s stress levels increased during surgery compared to the resting state, and the less experienced the surgeon, the longer the operation time, and the higher the blood loss, the stress levels of the surgeons increased [10, 11].

The stress levels and its effect of medical staff other than the surgeons are rarely studied [12-14]. Therefore, we aimed to analyze the intraoperative stress levels of the surgical nurses, which can help understand the hardships of the participating medical staff other than the surgeons and thus help in improving the teamwork of the surgical team. The intraoperative stress parameters of participating doctors and surgical nurses were analyzed using wearable devices. These real time data were used to analyze whether the stress patterns differed according to work experience and participating roles [12].

  1. Methods

    The Methods part is vague in the current version. For instance, what are the compositions of the 160 surgical stress records? Especially, the authors discussed the role differences later, but the detailed information about the role-relevant data was not clearly reported. The formulas calcuating concentration and tension were not clearly introduced. 

Response: We explained how the surgical stress records were obtained and the roles of each participating medical staff in the methods section. The composition of the records is clarified in Table 1 as the “enrolled cases in the present study”. The formulas for calculating concentration and tension are redefined to standardize the raw data obtained by the wearable device since the measurement units for the variables are different. We added further explanation on how concentration and tension were calculated, and provided further comments on each variable including the mid-range beta wave.

Page2, Material and Methods)

From June 2019 to December 2019, 160 surgical stress records, including intraoperative EEG signals and HRV data, were obtained during 40 spinal surgeries at a tertiary hospital. The operator, assistants, scrub nurses, and circulating nurses who participated in each surgery all wore the wearable device for data collection. Total 160 stress records were obtained from the participating medical staff.

The operator participated in all stages of the operation as the main surgeon, and the first assistant surgeon assisted the operator throughout the surgery. The scrub nurse prepared the instruments and materials and handed them directly to the surgeon. The circulating nurse checked for the overall process, prepared additional instruments not available in the surgical field, and aided the scrub nurses.

Page 4, Materials and methods)

Alpha, delta and theta waves appear when the person is relaxed, drowsy, idling or during meditation [21, 22]. Beta and gamma waves can be observed during active thinking tasks. Low beta waves are related to active, busy or anxious thinking and active con-centration. M-beta waves are associated with increases in energy, concentration, anxiety and performance. High beta waves are associated with significant stress, anxiety, paranoia, and high arousal [23-25]. Gamma waves appear when carrying out multiple cognitive functions [24]. Increased sensory-motor rhythm (SMR) is associated with a reduction in commission errors and improved cognitive function [26]. In regards to HRV-related measures, an increased low frequency/high frequency (LF/HF) ratio is associated with high stress levels [27].

Concentration is calculated by dividing the sum of SMR and M-beta waves by theta waves. Tension is calculated by dividing the H-beta wave by the alpha wave [28]. Because the collected data using the wearable device in this study had different measurement units, the company SOSO H&C suggested the following formula to calculate the concentration and tension [10, 11, 29-33]:

  1. Results

    The analysis methods were also not clearly introduced. In addition, since the numbers of the subjects were limited, the sample size was very small, in this case, the t-tests and ANOVAs may be not appropriate. Even though, the authors did not report the details of the statistical analyses.

Response: The number of stress records obtained in each group of the participating staff (operator, assistant, scrub nurse, circulating nurse) were 40, and therefore we performed ANOVA and t-test for comparison between the groups after checking for normality. For the result of stress-related parameters based on the work experience of scrub nurses (figure 4), the number of each group was small (n=10) and therefore Kruskal-Wallis test was used to compare the results.

  1. Dicussion

    The authors mentioned "team dynamics" many times when discussing the importance of the research topic and their findings. However, that is the definition of "team dynamics", which results supported the discussion of team dynamics?

Response: We intended to say that poor teamwork can lead to intraoperative stress in all medical staff participating in the surgery, however our team with experience over 6 months performing over 200 surgeries is unlikely to have stress due to poor teamwork. We agree with your opinion that our results do not show the level of teamwork (or team dynamics as we referred to previously). In fact, we thought our teamwork was not a factor which could affect the stress levels of the team members and therefore did not include this as a variable. We edited the methods section (page 3, line 97-99) to clarify our intent. We also edited the discussion section to reorganize the results, compare them with the previous studies and provide in depth discussion on the results of our study.